# Development of Professional Attributes through Integration of Science and Practice at First-Year Pharmacy Level

**DOI:** 10.3390/pharmacy9010004

**Published:** 2020-12-28

**Authors:** Theo J. Ryan, Sheila A. Ryder, Deirdre M. D’Arcy, John M. Quigley, Nyin N. Ng, Wuey Q. Ong, Zhong H. Tey, Máire O’Dwyer, John J. Walsh

**Affiliations:** School of Pharmacy and Pharmaceutical Sciences, The University of Dublin, Trinity College, Dublin, Ireland; sryder@tcd.ie (S.A.R.); ddarcy@tcd.ie (D.M.D.); jquigley@tcd.ie (J.M.Q.); NGNY@TCD.IE (N.N.N.); ONGW@TCD.IE (W.Q.O.); TEYZ@TCD.IE (Z.H.T.); MODWYER6@tcd.ie (M.O.); JJWALSH@tcd.ie (J.J.W.)

**Keywords:** integrated curriculum, integrative learning, curricular integration themes, competency framework, WHO analgesic ladder

## Abstract

The design, implementation and evaluation of a year 1 pharmacy-integrated learning component, using the World Health Organisation’s (WHO) analgesic ladder as a scaffold for case-based learning, is described. A novel aspect of the integrated component is the mapping of the cases to the national Core Competency Framework (CCF) for Pharmacists in Ireland and to the school’s own cross-cutting curricular integration themes. The integrated cases were student led and delivered through peer-to-peer teaching for 68 first-year pharmacy students. The integrated cases mapped strongly to three of the CCF’s domains, namely, *personal skills*, *organisation and management skills* and *supply of medicines*. With regard to the school’s curricular integrative themes, the cases mapped strongly to the curricular integration themes of *professionalism and communications*; *medicines sourcing*, *production and use*; and *safe and rational use of medicines*. Highlights from an anonymous online student survey were the recognition by students of the importance of core science knowledge for practice, the enabling of integrated learning and the suitability of the integrated component for entry-level. While a majority of students were found to favour individual work over group work, future iterations will need to consider a greater degree of group work with a view to reducing the volume of content and time required to complete the cases.

## 1. Introduction

Reference to curricular integration in pharmacy education has been notable in the literature, particularly in the last decade [1,2,3,4,5,6,7,8,9,10,11,12,13,14,15]. Accreditation bodies in many countries now require pharmacy programmes be integrated with clinical practice, thereby ensuring that the education and training of pharmacy students addresses the interdisciplinary skills required of the profession to improve delivery of care to patients. The need for higher education institutions delivering pharmacy programmes to meet the requirements in accreditation standards, particularly with respect to integration of science subjects with clinical practice, as well as competency attainment, has been described elsewhere, for example, in the United Kingdom by De Matteis et al. [15] and in the US by Jungnickel et al. [7]. Legislation enacted in Ireland in 2014 requires higher education institutions delivering pharmacy programmes to provide an accredited, five-year integrated pharmacy programme of education and training, leading to the award of M. Pharm. in preparation for professional practice [16]. The move to a five-year integrated pharmacy programme also provides for the integration of practice placements throughout the programme in years 2, 4, and 5. Prior to this, pharmacist qualification was attained following completion of a four-year bachelor’s degree with subsequent one-year professional internship (also leading to M. Pharm. qualification). The regulator of the pharmacy profession and the accrediting body for pharmacy education and training in Ireland is the Pharmaceutical Society of Ireland (PSI). The PSI define and describe the Core Competency Framework (CCF) for Pharmacists in Ireland [17] as a framework used to structure and guide continuing professional development (CPD) in pharmacy practice, which is based on the 2010 draft of the International Pharmaceutical Federation’s (FIP) Pharmacy Education Taskforce [18]. The CCF consists of six domains (Figure 1), subdivided into associated competencies and behaviours.

The five-year integrated programme of pharmacy education and training was implemented in the School of Pharmacy and Pharmaceutical Sciences (SoPPS) in the University of Dublin, Trinity College Dublin (TCD), Ireland in 2015. The first cohort of students to complete the five-year integrated pharmacy programme in the SoPPS graduated in Autumn 2020. The curriculum of the integrated programme is centred on five curricular integration themes (Figure 1) [19]. In order to support a student’s evolution to a competent practitioner, these curricular integration themes are directly aligned with the six domains of the CCF (Appendix A). To attain a flexible balance between disciplinary focus and curricular integration, all modules in the new programme have been assigned appropriate integration levels according to Harden [20], allowing for varying degrees of both integration and disciplinary depth. Furthermore, modules are aligned with the school’s five cross-cutting curricular integration themes, facilitating integrative learning throughout the programme [19].

Ideally, the pharmacy graduate from the integrated pharmacy programme will be a competent integrative learner who appropriately applies continuous learning to practice. However, it is recognised that integrative learning needs to be fostered [11]; therefore, it is essential to support and embed an integrative approach to learning in students at entry-level. This integrative learning needs to consider all CCF competencies, and, thus, all curricular integration themes should be clearly identifiable in year 1 of the programme. It has been suggested that integration should be considered from the beginning of the pharmacy programme using a spiral integration approach [13], but, to our knowledge, there is a shortfall in the literature of entry-level competency-based integrated elements in pharmacy curricula [4,21,22]. In many of the instances in the literature where competency development is addressed at pharmacy entry level, it is invariably aligned to programme accreditation standards rather than to a professional competency framework for practising pharmacists [2,10,15,23]. Therefore, this was identified as a requirement within our programme, and, furthermore, it would help to address the paucity of evidence in the literature supporting competency-based integrative learning at entry-level in pharmacy curricula. 

The aim of this paper is to describe the design, implementation and evaluation of an entry-level (year 1) competency-focussed case-based integrated component, in a five-year integrated pharmacy programme in Ireland. 

The following are the objectives: 

(i) To describe the design and implementation of the case-based integrated component which was centred on the WHO analgesic ladder for a first-year pharmacy class [24];

(ii) To evaluate alignment of the integrated component with pharmacy practitioner competencies, specifically by mapping the integrated case-based exercises to the CCF and the school’s curricular integration themes;

(iii) To evaluate students’ perceptions of the integrated component exercises on their learning and competency development through an online anonymous survey.

## 2. Methods

### 2.1. Design and Implementation of the Integrated Component

Six academics from the SoPPS (J.W., S.R., M.O’D., T.R., D.D. and J.Q.), representing the specialities of chemistry, pharmacology and pharmacy practice, collaboratively designed 68 integrated pain-case exercises, one for each student in the class, using the WHO analgesic ladder as a scaffold for integrated learning [24]. The approach taken to integration within the case-based exercises centred on the school’s cross-cutting curricular integration themes and the domains of the CCF (Figure 1). The integrated component was to be completed in advance of the students undertaking a two-day community pharmacy experiential placement. Pain was selected as the topic as it was considered that even entry-level students commencing the programme directly from second-level education (i.e., most students) would be familiar with the concept, causes and symptoms of pain. Furthermore, the medications used in the treatment of pain are a focus across year 1 modules (Appendix B). Using a broad topic such as pain as a cognitive hook would allow several areas of knowledge across the curriculum to be drawn upon in an integrative manner, from origin of drug substances to the appropriate prescription and supply to the patient. 

The 68 integrated pain-case exercises were grouped under six categories: 1. pathophysiology; 2. paracetamol; 3. capsaicin; 4. salicylates; 5. other nonsteroidal anti-inflammatories (NSAIDs); and 6. opioids. The content for the case exercises within each category were based on the WHO’s analgesic ladder and were developed starting from their chemical/pharmacological details and moving towards the clinical and practice considerations of the drug(s) in question. An example of a pain-case exercise is provided in Appendix C. Each student was required to prepare a 3 min presentation on their assigned case. Prior to each student being assigned their individual case, an academic member of staff (J.W.) briefed the students on the integration philosophy of the school, highlighting the school’s curricular integration themes and the CCF. Leaders, selected from the class, were assigned to each of the pain-case categories. Each category leader had responsibility for collating the students’ presentations within that category. Before circulation of the student presentations to the class, the academic staff critiqued each presentation for accuracy of information. Where inaccuracies in content were evident, the students were notified and required to submit a corrected version prior to the students’ presentations, which were delivered over two three-hour sessions. The integrated component consisted of guided peer-to-peer pain-case presentations, where students developed and delivered their case individually. Student assessment by the academic staff was based in part on each student’s presentation content and communication skills and in part on the peer-to-peer learning by means of a crossword test, containing 30 clues. Assessment of the students’ learning is outside the scope of this paper.

### 2.2. Mapping of the Integrated Component with Practitioner Competencies

To evaluate alignment of the integrated component pain cases with practitioner competencies, each case was mapped by the academic team to the following:

(i) The domains of the CCF;

(ii)The school’s curricular integration themes. 

When mapping the cases to the six domains of the CCF, the content of each case and the tasks required of the students were considered against the competencies and behaviours pertaining to each domain of the CCF [17]. Similarly, when mapping to the school’s curricular integration themes, cases were considered against the descriptors of the curricular integration themes (Appendix A) [19]. As the school’s curricular integration themes are broader in scope than the CCF domains, on occasion, a more restricted interpretation of the definition of the integration theme was employed with a view to better discriminative ability when mapping. The mapping task was undertaken by three academic staff (T.R., D.D. and J.W.).

### 2.3. Students’ Perceptions of the Integrated Component Exercises 

Ethics approval was granted by the SoPPS’ Research Ethics Committee, Trinity College Dublin. An anonymous online survey (Appendix D) was made available to all students through SurveyMonkey^®^ after completion of the students’ pain-case presentations and their two-day experiential learning placement to ascertain student perceptions of the integrated component on their competency development and learning experience.

Regarding competency development, students were asked how the integrated component improved different skill subsets and rated their agreement to statements on a five-point Likert scale, ranging from “strongly agree” to “strongly disagree”. The skills addressed in the questionnaire were the following: teamwork; presentation and communication; critical thinking; research; working independently; and time management. 

Similarly, students were asked how the integrated component impacted on their year 1 learning experience and rated their agreement to statements on the same five-point Likert scale. This element of the questionnaire addressed student perceptions of the relevance of the integrated component cases to positive learning, integration of coursework, experiential learning, year 1 suitability, confidence building and the importance of science subject for practice.

Students were asked about preference for groupwork compared to individual work. The final part of the questionnaire consisted of semistructured components enabling students to provide free text comment in answer to specific questions on their perceptions of the most beneficial aspect of the exercise and suggestions for subsequent improvement.

### 2.4. Statistical Analysis

Open text content data from the survey were analysed using QDA Miner Light Version 2.0.2 and SPSS Version 24. Quantitative data from the student survey were analysed using SPSS Version 24, with descriptive data being presented as percentages and corresponding n numbers, and continuous variables as means and corresponding range.

## 3. Results

### 3.1. Mapping of the Integrated Component with Practitioner Competencies

#### 3.1.1. Mapping of the Integrated Component to the Core Competency Framework (CCF) for Pharmacists in Ireland 

The results of mapping the 68 integrated pain cases to the CCF are illustrated in Figure 2. When mapping the pain cases to the practitioner competencies of the CCF, all cases addressed the domains of *personal skills* and *organisation and management skills*, based on the individual nature of the work required of each student. The domains of the CCF to which the pain cases were next most frequently mapped were *supply of medicines* at 85% (n = 58), *safe and rational use of medicines* at 54% (n = 37) and *public health* at 54% (n = 37). The domain to which the pain cases mapped the least was *professional practice*, at 31% (n = 21). 

#### 3.1.2. Mapping of the Integrated Component to the School’s Curricular Integration Themes

The mapping of the 68 integrated pain cases to the school’s curricular integration themes is illustrated in Figure 3. All 68 pain cases mapped to the curricular integration theme of *professionalism and communications*, based more on the element of communication rather than professionalism, as all students were required to develop and present their own case to their peers. The curricular integration themes of *medicines sourcing, production and use* (85%; n = 58) and *safe and rational use of medicines* (54%; n = 37) were well represented within the cases, while the curricular integration theme that was least incorporated into the case-based exercises was *pathologies*, *patients and populations*, being only evident in 9% (n = 14) of the cases. It should be noted that while pain is obviously related to pathology, and on that basis all cases could have been mapped to that theme, for increased discriminative ability in mapping, only cases that represented commonly encountered patient groups in the community and public health issues were mapped to the theme of *pathologies*, *patients and populations*.

### 3.2. Evaluation of Students’ Perceptions of the Integrated Component Exercise

In total, 51 students out of the class of 68 responded to the online evaluation questionnaire, giving an overall response rate of 75% for the survey. Some students chose not to answer all questions, or all parts of a question, thereby resulting in variability of the response rates from question to question.

#### 3.2.1. Evaluation of Students’ Perceptions of the Integrated Component Exercises on Competency Development

Students’ perceptions of the impact of the integrated component on competency-focussed skills were also evaluated (Figure 4). The different parts of this question were answered by between 41 to 43 students (82–86% response rate).

The students were particularly satisfied that following the integrated component, they considered that their ability to develop and deliver a presentation and to conduct research had improved “a lot” or “somewhat” (95%; n = 40) in both instances. With cases assigned individually to students, the skill of “teamwork” was identified as being improved upon by only 34% of respondents.

#### 3.2.2. Evaluation of Students’ Perceptions of the Integrated Component Exercises on the Learning Experience

Student perceptions of the impact of the integrated component pain cases on their learning experience are presented in Figure 5. This question covered various aspects spanning the positivity of the learning experience, integration of coursework, relevance to experiential learning, suitability for year 1, confidence building through presentations, and the importance of science subjects for practice. The question was answered by 42 students, giving a response rate of 62%.

In total, 91 (n = 38) of students either “strongly agreed” or “agreed” that the sciences are important to practice. Similarly, 91 (n = 38) of students either “strongly agreed” or “agreed” that the integrated component was a positive learning experience and 93 (n = 39) either “strongly agreed” or “agreed” that the integrated component was pitched at a suitable level for year 1 students. Regarding integration of learning, 83 (n = 35) of students either “strongly agreed” or “agreed” that the integrated component had helped them to integrate their learning across different modules of the programme in year 1. Overall, 57 (n = 24) of students believed that the integrated component was relevant to their two-day experiential learning placement in community pharmacy practice.

#### 3.2.3. Group Work versus Individual Work 

Students were asked if they would prefer in future iterations to undertake the case-based exercises in groups or individually, with 41 students in total responding to the question. Most students (68; n = 28) stated that they would prefer to work individually, with 32 (n = 13) favouring group-based activity. For those students who responded that they preferred an individual approach, the main themes identified from content analysis of student’s free-text answers showed that students believed that they would have improved learning as an individual and reduced preparatory time for the cases.

“It is better for everyone to do their own work and then bring it all together at the end, as many people find it difficult to work to their full potential in groups,” Student #36.

“I felt that working in groups could slow down the research rather than speeding it up,” Student #48.

For those students who responded that they would have preferred to complete the activity as a group-based activity, the main reasons cited were the possibility of improving their teamwork skills, a reduction in time to complete the activity and overall expected improvement in their learning by working in groups.

“Doing work in groups helps to prepare you for your career as you will be dealing with others and improves teamwork skills, communication and cooperation.” Student #35.

“I feel doing the case study in groups would have resulted in more information and also given me more confidence when presenting.” Student #26.

#### 3.2.4. Student Perceptions Regarding Positive Aspects of the Integrated Component

Students were asked as to what they considered the best aspect of the integrative component, with 40 students answering this question. Three main themes emerged from the content data analysis of the 40 free-text responses, which accounted for 78 (n = 31) of the responses, which were the following:

(i) Development of the ability to integrate learning across the programme:

“I actually got to see where everything came together from different modules. Some parts were so interesting and I really got to learn about parts I had seen in my work placement.” Student #37.

“Seeing everything that you learn in lectures coming into practice. Sometimes in lectures it is hard to imagine how you will use some of the content in real life—however the case study showed this,” Student #46.

(ii) Development of presentation skills:

“I found presenting my topic very helpful as it gave me confidence with regards to imparting knowledge to my peers.” Student #40.

(iii) Development of knowledge of the subject:

“Learning useful information and seeing how it this knowledge could be put into practice,” Student #23.

These three themes accounted for 78 of the students’ responses as to what worked best for the integrative component. A smaller number considered the crossword-based assessment (5; n = 2) to have been the best aspect, while others considered having the opportunity to learn from their peers (10; n = 4).

#### 3.2.5. Student Recommendations for Future Changes to the Integrated Component

When asked if such case-based integrative components should be utilised for later years of the programme, 93 of respondents (n = 42) agreed that it should be continued. Content data analysis of the students’ free-text responses supporting future use of the integrative component revealed two main themes:

(i) An enhanced way of learning:

“It’s a good, interactive way of learning,” Student #19.

“Independent research on a given topic encourages the student to find out information for themselves rather than being told what to say or learn. I found it helped me to remember the information more efficiently when I found it for myself.” Student #9.

(ii) Enabling integration of coursework:

“I believe that continuous integration between modules throughout all years of the course would be very beneficial as it allows us, as students, to make links between our learning and therefore understand the relevance of learning what we are.” Student #40.

Improvement in presentation skills was referred to frequently in support of retaining the integrated component for later years.

Students were asked what could be done to improve the integrated component (n = 43). Content data analysis of the students’ free-text responses highlighted that the main issues concerned the time requirement involved case complexity and the large volume of information disseminated through the multiple student presentations.

“Some of the presentations were very long with too much information in them to comprehend during the allocated time. I think it would be better to keep the performances to the maximum of three minutes.” Student #18.

Coupled to these observations was the recommendation by some students to switch from individual presentations to group work, with a view to reducing the time taken for presentations and the volume of information delivered.

## 4. Discussion

### 4.1. Design and Development of the Integrated Component

In this paper, we describe the design, implementation and evaluation of a new integrated component using case-based learning for first-year pharmacy students in an integrated pharmacy programme in a university in Ireland. This study provided students with a novel opportunity to integrate material throughout the first-year course from basic pharmaceutical sciences to pharmacy practice. The enhanced requirements in Ireland for integration of pharmacy practice and science in pharmacy programmes place significant time and resource demands on academics and challenge us to create new approaches to integrate and understand students’ perceptions and experience of integration.

Despite the academic team being experienced in teaching at pharmacy undergraduate level, developing theme-based integrative case studies was a novel method of teaching, enabling increased collaboration across disciplines. These increased collaboration across disciplines, and new ways of working have also been recently experienced and described by other schools of pharmacy [15]. Pain has been described by Husband et al. as “a cognitive hook”, tapping into what students may already know, and, thus, it is particularly suitable as a topic for year 1 students [5]. Furthermore, medications used in the treatment of pain are of focus in the modules in year 1 in the SoPPS (Appendix B). For example, students synthesise paracetamol and aspirin within the chemistry practical classes, students develop their clinical skills relating to the appropriate management of headache in the practice of pharmacy module and the WHO stepped approach to pain management is appraised, also within the practice of pharmacy module.

### 4.2. Mapping of the Integrated Component with Pharmacy Practitioner Competencies

Mapping of the integrated component exercises to the competencies required of pharmacy practitioners in Ireland was undertaken to support the provision of competency-focussed integrative learning content early in the curriculum. Students need to attain competence in the CCF on graduation, implicitly conferring value on the alignment of learning with the CCF from year 1; on the other hand, alignment of learning directly with the school’s cross-cutting curricular integration themes facilitates intra- and inter-module curricular integration, across all years of the programme. It is clear from Figure 2 and Figure 3 that all domains of the CCF, and all curricular integration themes, can be mapped to the case-based exercises, illustrating the successful application of the concept of a competency-focussed integrative learning component. The lower frequency of mapping to the *professional practice* domain and *pathologies, patients and populations* theme is not surprising, given the entry-level knowledge base of the student cohort.

### 4.3. Student Feedback on the Integrated Component 

Our findings revealed that students considered that their ability to develop and deliver a presentation and to conduct research had improved through the case studies, thereby helping to address the *personal skills* and the *public health* domains of the CCF at an early stage of student development, respectively. Importantly, over nine in ten students indicated that the case studies improved their presentation skills, including their oral communication skills. This is an important skill for developing pharmacy students and future pharmacists, and students previously had limited opportunities to develop these skills in the first year of the pharmacy programme. Students also felt that the integrated component helped them to develop their critical thinking skills (aligned to the *personal skills* domain of the CCF), and this was similar to findings by Kullgren et al. in the delivery of a similar integrated course in pain management [10]. Only one-third of students indicated that the integrated component improved their teamwork skills, which is unsurprising given the fact that the students completed the cases individually. Furthermore, in terms of skill development aligned with the CCF, there was scope to enhance the opportunity to improve teamwork skills, (a competency in the *personal skills* domain of the CCF). On reflection, after reviewing the student feedback and case studies roll out, the academic staff decided to take this feedback into account, and it was decided that in subsequent years, the students should complete the cases in pairs. This amendment also helps to address the issues associated with the volume of information and time requirements for presentations identified by students and staff.

Student’s reported impressions of the integrated component on their learning experience were positive, with 91% “strongly agreeing” or “agreeing” that the cases studies were a positive learning experience. This correlates well with findings by De Matteis et al. from a similar entry-level integrated component, where 89 of students reported that the integration had aided their learning. As regards integration as a pedagogical approach, over eight in ten students “agreed” or “strongly agreed” that the integrated component enabled them to integrate learning across the course, a finding strongly supported by Kullgren et al. [10].

The perceived low impact of the integrated component on students’ experiential learning, coupled with the lower frequency of mapping to the *professional practice* domain, suggests scope for increased signposting of relevance of the content to practice. This could be reflected in experiential learning placement exercises or briefings, and cross-referencing with relevant content in modules in later years.

It has been noted that assessments of integrated curriculum components are lacking, and also that assessment of the students undertaking the components (i.e., the actual student assessment) has pitfalls such as overlap in examination questions and content redundancy [9]. The exercise presented in this study represents a significant academic time commitment from staff in addition to students. However, the collaboration from the cross-disciplinary team in case design served to reduce the content overlap, and basing the assessment on the student presentations and crossword, avoided overlap and redundancy in assessment.

### 4.4. Strengths

To our knowledge, this is the first study describing the design, implementation and evaluation of an entry-level integrated element in a pharmacy programme in Ireland that is centred on a professional competency framework for pharmacists and uses cross-cutting curricular integration themes.

### 4.5. Limitations

In this paper we have described a single approach in a single institution in Ireland; thus, the results may not be generalizable to other institute or countries. It is planned to further customise the integrated component, to improve it based on student and staff feedback and to use it as an approach for later years of the programme.

## 5. Conclusions

An integrated component has been successfully developed and implemented for entry-level (year 1) pharmacy students, using the WHO’s analgesic ladder for case-based learning. A novel aspect of this integrative component is the mapping of the integrative pain cases to a national competency framework for practising pharmacists in Ireland and to the school’s own curricular integration themes. The mapping exercise has shown that all competency domains and curricular themes could be mapped to the integrative component cases. The lowest frequency of competency mapping was to the *professional practice* domain.

Highlights from the student evaluation were the importance of science subjects for practice, the integration of learning and suitability for entry-level students. While most students were found to favour individual versus group work, future iterations of the integrative component will need to consider a greater degree of group work with a view to reducing the volume of content and time required to complete the student cases. The lowest perceived area for impact of the exercise on student learning related to their practice placement. This finding, combined with the outcome of the mapping process, illustrates how evaluation of the exercise valuably highlights the scope to further scaffold relevant campus-based learning to experiential practice placement learning. We believe that the work presented provides a basis for other schools of pharmacy to integrate science with practice across multiple disciplines, whether using integration themes in a cross-cutting approach to curricular integration or using professional competency frameworks.

## Figures and Tables

**Figure 1 pharmacy-09-00004-f001:**
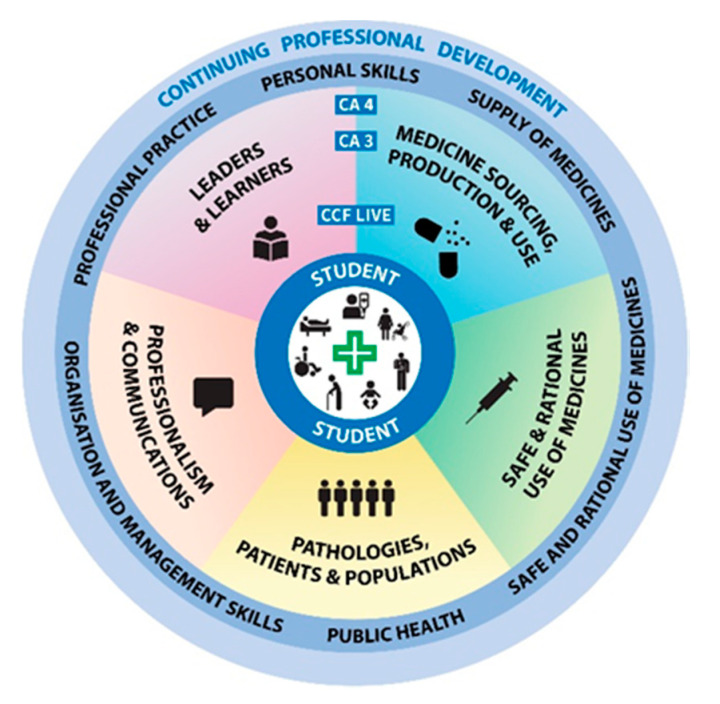
Visual presentation of the integrated pharmacy curriculum, encompassing the six professional domains and five curricular themes. The six domains of the Pharmaceutical Society of Ireland (PSI)’s Core Competency Framework (CCF) for Pharmacists in Ireland (detailed in the second ring from outside), based on the International Pharmaceutical Federation’s (FIP) global competency framework, encompassing the school’s five cross-cutting curricular integration themes (detailed in the third ring, colour coded for each theme).

**Figure 2 pharmacy-09-00004-f002:**
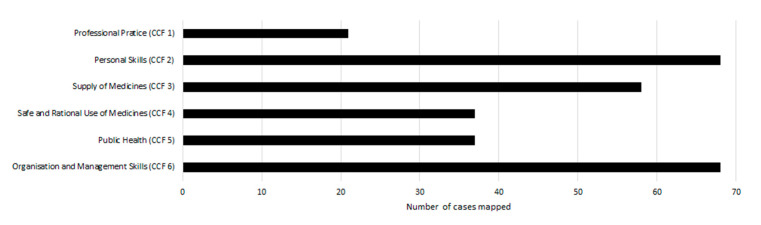
Integrated component cases mapped to the six domains of the Core Competency Framework (CCF) for Pharmacists in Ireland (n = 68).

**Figure 3 pharmacy-09-00004-f003:**
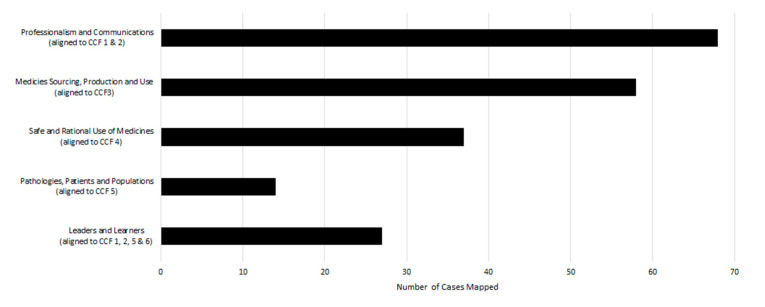
Integrated component cases mapped to the school’s curricular integration themes (n = 68).

**Figure 4 pharmacy-09-00004-f004:**
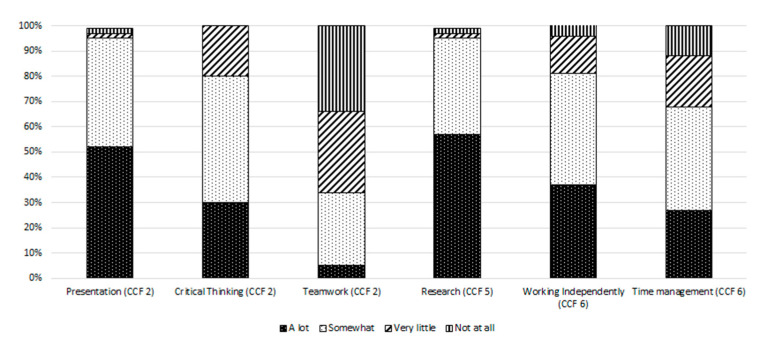
Students’ perceptions of the degree to which the integrated component positively impacted the various skills as shown, and as aligned to the relevant domain (1–6) of the Core Competency Framework (CCF) for Pharmacists in Ireland.

**Figure 5 pharmacy-09-00004-f005:**
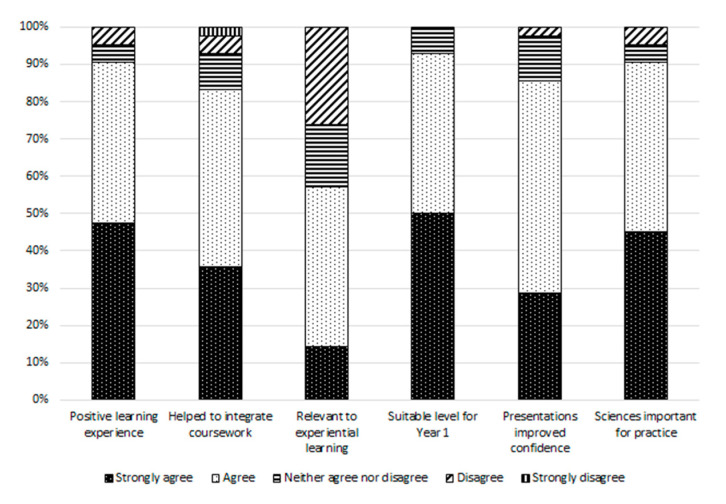
Student perceptions of the impact of the integrated component cases on their learning experience (n = 42).

## Data Availability

Data is available from the author upon request.

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
