# Peer review of "Development of Professional Attributes through Integration of Science and Practice at First-Year Pharmacy Level"

_pharmacy, 2020, doi:10.3390/pharmacy9010004_

Round 1
Reviewer 1 Report
Overall, I think the concepts presented in the paper are important and contribute to the literature in how we incorporate competency based education into pharmacy education internationally. I applaud the authors for working across departments to integrate various components of Year 1 in the program. The introduction set things up nicely and there was a clear purpose to the paper. I will outline areas for improvement below. I think with minor changes the paper could really be strengthened to make it easier for the reader to digest.
- Unfortunately there are many different ways to define 'integration' when discussing curriculum. Based on your reference list you seem to reference both true curricular integration (i.e. horizontal, disease states modules, etc.) and also integrating practice experiences into the curriculum. I would be more clear about what you are defining as 'integration' within the context of your work.
- Please evaluate a different way to present your survey response rate in the methods. It is very confusing with each question being answered by a different number of students. In addition, in one section of the paper the authors state that '41-43' students completed the survey for a response rate of 82-86% and in another area 42 students was a response rate of 62%. There may need to be some language added prior to presenting the survey results that explain if it was administered in parts, if applicable and why there are differences for each question.
- In addition most figures represented in the paper have an (n=) in the caption except for Figure 5. I would recommend adding one for clarity.
- The qualitative comments embedded within the text of the paper make it harder to follow the major themes gathered from free text responses. Consider if presenting summarized qualitative comments in a table might present the data more concisely and not break up the narrative of the results section.
- In section 3.2.5 there is a lot of overlap in comment themes from the 7% of students who would not retain the activity and what other students said about ways to improve the activity in the future. Can you be more concise with those descriptions to prevent repeating information?
- Sections 4.1 and 4.2 of the discussion section should be reviewed. Many elements in these two sections referenced methods that had already been presented and were not drawing conclusions based on the findings. Certain elements of these sections may need to be removed for brevity.
The paper does require minor editing for grammatical correctness and spelling. I noticed a few instances of dropped words from a sentence or spelling errors.

Reviewer 2 Report
Thank you for the opportunity of reviewing your interesting article. It addresses a topic which is within the journal s scope and uses relevant literature to perform the content analysis. Some specific issues: Please, consider enhancing the final part by providing more conclusions/implications/further directions of research etc. Appendix 1, 2, and 3 should be after section References.Please consider verifying the language, there are some grammar/writing errors, for example:
- thereby ensuring that the education and training of pharmacy students addresses - ADRESS
- as well as competency attainment, have been described - HAS BEEN DESCRIBED
- but to our knowledge there is a shortfall in the literature of entry-level competency-based integrated - COMMA IS MISSING
- for practicing pharmacists - PRACTISING
- campus-based leaning - LEARNING...
Reviewer 3 Report
It is not very often that one comes across a description of integrated instruction at the year one level in the pharmacy curriculum. Very interesting design and description. Hopefully can be adapted by other faculties once published.
I would suggest including an example of the pain case exercise so the reader can get a sense of the content/style.
